# Experiences during the COVID-19 Pandemic among People with Inflammatory Arthritis: “Reopening of Society Is Harder than Lock-Down”—A Qualitative Interview Study

**DOI:** 10.3390/vaccines10070982

**Published:** 2022-06-21

**Authors:** Lene Dahl Lund, Mette Margrethe Løwe, Oliver Hendricks, Karen Schreiber, Bente Glintborg, Randi Petersen, Christiane Plischke, Willy Fick, Jette Primdahl

**Affiliations:** 1University College UC SYD, Research Center of Applied Health Science, 6705 Esbjerg, Denmark; mette.lowe@rsyd.dk; 2Danish Hospital for Rheumatic Diseases, University Hospital of Southern Denmark, 6400 Sønderborg, Denmark; ohendricks@danskgigthospital.dk (O.H.); kschreiber@danskgigthospital.dk (K.S.); rpetersen@private.dk (R.P.); christiane-dk@live.dk (C.P.); willy.fick@hotmail.com (W.F.); jprimdahl@danskgigthospital.dk (J.P.); 3Department of Regional Health Research, University of Southern Denmark, 5000 Odense, Denmark; 4Copenhagen Center for Arthritis Research (COPECARE), Center for Rheumatology and Spine Diseases, Centre of Head and Orthopedics, Copenhagen University Hospital Rigshospitalet, 2600 Glostrup, Denmark; glintborg@dadlnet.dk; 5Department of Clinical Medicine, Faculty of Health and Medical Sciences, University of Copenhagen, 2200 Copenhagen, Denmark; 6Sygehus Sønderjylland, University Hospital of Southern Denmark, 6200 Aabenraa, Denmark

**Keywords:** qualitative research methods, COVID-19, vaccination, inflammatory arthritis, DMARDs

## Abstract

People with inflammatory arthritis (IA) treated with immunosuppressive disease-modifying anti-rheumatic drugs (DMARDs) were initially considered to have an increased risk of severe illness from the SARS-CoV-2 virus compared to the general population. The aim of this study was to explore how people with IA experienced restrictions during the pandemic and the possible impact of vaccination on their protection against COVID-19 and their everyday lives. Nineteen people with IA were interviewed in May–August 2021; shortly thereafter they were enrolled in the Danish national COVID-19 vaccination programme. Concurrently, society gradually reopened after a national complete lockdown. The analysis was inspired by inductive qualitative content analysis. Participants expressed a lack of targeted information on the specific risk associated with IA if they contracted COVID-19. They had to define their own level of daily-life restrictions to protect themselves and their families. They were impacted by inconsistent announcements by the authorities, and some expressed concerns regarding the potential influence of DMARDs on vaccine effectiveness. A societal spirit of being “in this together” emerged through the lockdown, and some were concerned that the reduced level of restrictions in the reopened society would put them at higher risk of a COVID-19 infection and force them to continue self-isolating.

## 1. Introduction

In December 2019, the first outbreak of the SARS-CoV-2 virus was reported in Wuhan, China [1,2]. The highly contagious virus quickly spread to other countries, and in Denmark the first case of the disease caused by SARS-CoV-2 (COVID-19) was reported on 27 February 2020 [3]. On 11 March 2020, the World Health Organization declared COVID-19 to be a pandemic [4], and on the same day the Danish government introduced a nationwide lockdown. Universities, schools, training facilities, restaurants, shopping centres, etc., were closed and people were recommended to work from home if possible [5]. Society reopened after the first wave of COVID-19 during summer 2020; however, a second wave and another lockdown followed between December 2020 and April 2021. In the following months, there was a gradual reopening, where most restrictions were lifted, such as the requirement of social distancing [6].

The first vaccinations against SARS-CoV-2 were offered to the most vulnerable citizens in Denmark in December 2020. Due to the scarcity of vaccines at that time, priority was based on factors such as higher age and estimated risk of a more severe COVID-19 infection, including those with chronic illnesses, such as chronic inflammatory arthritis (IA). Contrary to initial expectations, studies did not confirm that those with IA treated with immunosuppressive disease-modifying anti-rheumatic drugs (DMARDs) had an increased risk of contracting COVID-19 or higher mortality compared to the general population [7,8,9]. Thus, in March 2021, the Danish vaccination programme was adjusted, in accordance with recently published evidence that the most significant risk factor for contracting a serious course of COVID-19 was high age, given that 87% of all COVID-19-related deaths were among people aged 70+. At that time, the SARS-CoV-2 Delta variant was driving infections in Denmark, and IA was no longer considered a particular risk [10].

Studies have found that there was lower psychological wellbeing and potentially higher rates of anxiety and depression in the general population during the first lockdown compared to the time before the COVID-19 pandemic [11,12,13]. Female gender, poor self-rated health, and being a relative of a person who had COVID-19 increased the risk for negative emotional reactions [12]. Mancuso et al. reported that coping with challenges due to the pandemic both directly and indirectly worsened symptoms in patients with IA [11]. A Danish nationwide study found that among 12,789 persons with IA, 70% were afraid to contract COVID-19, 75% thought they had high risk of contracting the infection, and up to 48% isolated themselves more than others their age in March 2020 [14]. However, a follow-up study found that the proportion who considered themselves to be of high risk decreased to 63% in June 2020 [15].

The final rollout of the vaccination programme and the reopening of society in Denmark took place during the summer of 2021. This meant that many, mainly younger, people with IA had not received the first dose of the SARS-CoV-2 vaccine when the assembly ban and distancing requirements were lifted. At that time, no medical treatment was available or approved by the Danish Medicines Agency. Furthermore, other studies have raised concerns as to whether people with IA achieve adequate response to the vaccination when treated with DMARDs [16,17]. Therefore, this study aims to explore how people with IA experienced both the lockdown and the following reopening of society, and their thoughts about the impact the SARS-CoV-2 vaccination would have on their everyday lives and protection against COVID-19.

## 2. Materials and Methods

### 2.1. Setting and Participants

Participants were recruited through the DECODIR study “Detection of SARS CoV-2 antibodies in Danish Inflammatory Rheumatic Outpatients—An observational cohort study” [17]. The DECODIR study aimed to investigate the immune response to and side effects of COVID-19 vaccines given to people with IA and compare immune response between patients on and not on DMARD treatment [17].

We invited 19 patients who were already selected to participate in DECODIR to take part in the present study. They all agreed to participate. We used a purposeful sampling strategy aiming for maximum variation in relation to age, gender, civil status, diagnosis, and treatment with DMARDs. We also included two participants who were invited to take part in the DECODIR study, but who had decided they did not want the vaccination.

### 2.2. Interviews

An interview guide was developed by the research team, which included two patient research partners. The interview guide was based on previous research in relation to living with IA during the COVID-19 pandemic [11,12,15]. The interview guide focused on two research questions in relation to the aim of the study: (1) How did persons with IA experience restrictions during the COVID-19 pandemic? and (2) What are their thoughts about the possible impact of a SARS-CoV-2 vaccination on their protection against COVID-19 and their everyday lives?

The interview guide consisted of 5 open-ended questions, as shown in Table 1, followed by specific probes.

Authors 1, 2, and 6 conducted the interviews by phone or online via the online video platform Teams, depending on the participants’ preferences. This was considered appropriate due to the risk of spreading COVID-19. We also had the opportunity to conduct interviews at the hospital as an extension of a clinical consultation, which one participant preferred. Each interview was opened with a briefing on the agenda and the purpose of the interview. The interviewers placed great emphasis on showing empathy and establishing a trusting atmosphere, e.g., using active listening and providing space for reflections throughout the interview [18,19].

The interviews were conducted from May to August 2021, shortly after the participants had, or were offered, their first dose of the COVID-19 vaccination. After the first interviews, the researchers met to discuss and align their interview style and adjust the interview guide.

The interviews were audio-recorded and transcribed verbatim in the software programme NVivo, version 12, by a student assistant with experience in transcribing in NVivo.

### 2.3. Analysis

The analysis was inspired by the inductive qualitative content analysis, as described by Graneheim and Lundman [20]. This approach allowed us to work with both the manifest and the latent dimensions of the material, and thereby fulfil the aim of the study. The manifest content dealt with the obvious components, whereas the latent level involved interpretation of the underlying meaning of the text. The steps in the analysis are described in Table 2.

### 2.4. Patient Research Partners

Two patient research partners (PRPs) played a central role throughout the study. They were involved in the development of the study protocol and the interview guide. The PRPs were actively involved in all research group meetings and contributed to the data analysis with their experiences of living with an inflammatory rheumatic disease. Both PRPs had been diagnosed with IA and were connected with the Danish Hospital for Rheumatic Diseases as outpatients, as were the participants in the study. By way of a workshop initiated by the researchers, they were also engaged in the interpretation of findings and, later, as co-authors on the manuscript.

### 2.5. Ethical Considerations

The Regional Committees on Health Research Ethics for Southern Denmark were contacted but waived the need for a formal approval (Ref. ID 20212000-91). Oral and written consent were acquired before participation, and the participants were informed about their right to withdraw from the study at any time and that non-participation would not affect their treatment. Data were stored and analysed in OPEN Analysis, a safe environment that complies with the current Danish Data Protection Law and the European General Data Protection Regulations [21]. Storage and management of the data were registered under the Danish Data Protection Agency. Rules of confidentiality were observed, and no names or other sensitive personal information are reported in the study. To protect participants’ anonymity, their age is only referred to in 10-year intervals.

## 3. Results

In total, 19 participants were included. The participants’ ages ranged from 21 to 64 years, with a median of 50 years. Seven were male, 13 lived with a partner, and six lived alone. They were all diagnosed with IA: Three with psoriatic arthritis (PSA), three with spondyloarthropathies (SpA), nine with rheumatoid arthritis (RA), two with Morbus–Bechterew (MB), and two with juvenile inflammatory arthritis (JIA). Seven were treated with conventional DMARDs and seven were treated with biological DMARDs, as illustrated in Table 3.

As displayed in Table 4, five themes were derived from the analysis. In the following sections, each of the themes is described and supported by selected quotations. The participants are referred to by the numbers displayed in Table 3.

### 3.1. Changing and Divergent Information

A majority of the participants wished for targeted information about IA and COVID-19. They believed they were at particular risk of a serious disease course if they contracted COVID-19, due to their treatment with immunosuppressive drugs. The participants experienced that information from the health authorities on COVID-19 had changed during the first lockdown, before the vaccines were released, and it caused a lot of doubt and concerns.

*Initially it was stated that we were vulnerable (…) I guess we were until around New Year’s or something, then a slide started happening, right? The Board of Health began to assess it differently. Then all of a sudden, we weren’t so much at risk (…)* (2)

Several participants perceived the information they had received as divergent.

*Now I don’t know what to believe (…). Because I went to a consultation at the arthritis hospital. There I asked again and they said; no, you are no more vulnerable than others.* (5)

The participants searched for information about COVID-19 and IA. They all cited doctors and nurses at the hospital as essential sources of valid knowledge. The Internet and Facebook, and especially The Danish Rheumatism Association webpage [22], were important sources for information. In addition, the participants’ relatives and those in their social networks were common sources of information about COVID-19.

*Well, it’s both the internet and my dad, he’s very keen on us having to take care of ourselves. So, he sends links to various things and cases, investigations, and studies (…) Have also talked a lot with my doctors at the hospital (for Rheumatic Diseases) about it.* (13)

Some participants found that there was an overload of general information given to the public, whereas there was a lack of targeted information regarding their risk as IA sufferers and because they were being treated with DMARDs. Most of the participants did not want to burden their general practitioner (GP) with a lot of questions. One participant highlighted a livestream on the Danish Rheumatism Association Facebook page as a turning point in her knowledge, and another participant suggested that the hospital could create a newsletter.

### 3.2. Individual Interpretation of Own Risk

The participants had to conduct their own interpretation of their risk and consequences, based on existing knowledge and continuously changing information, from not only the authorities, but also health professionals, social media, etc., partly illustrating the rapid change in evidence. They had to decide on their own level of daily life restrictions—a task some found very difficult, whereas others were familiar with self-imposed precautions.

*(…) But then it’s up to me, suddenly, to assess; do I think I’m at risk or do I think I’m not at risk? (…) As a rheumatoid arthritis patient, you already take care of yourself. If you saw someone coughing or sneezing a little further on, you went in a big arc around them, because you had been told that the medicine lowers your immune system, that you are more susceptible.* (2)

Several participants expressed that they had felt alone in assessing their own risk. They felt it was their own responsibility to judge the consequences a potential infection could have on them. One participant felt “abnormal,” perhaps even labelled, as her friends were not concerned about contracting COVID-19 at all.

Some participants reacted pragmatically to the uncertainty and doubts, emphasising that arthritis is not fatal, like cancer can be. Nevertheless, they wished for more information about their risk and vulnerability. Overall, the most common concerns were whether they had an increased risk of contracting COVID-19, whether they would experience more severe symptoms than others, and whether a COVID-19 infection could lead to exacerbation of their IA:

*Now, if I get infected with COVID-19, will it affect my arthritis, so that it gets a lot worse?* (3)

The participants also reflected on whether they would need to pause their DMARD treatment if they contracted COVID-19 and therefore would risk complications, such as increased pain and/or impairments.

*(…) and then when I get sick (with COVID-19), I’m not allowed to take my medication, then some other ailments pop up that the medication dampens. So it wasn’t just that I could possibly risk getting Corona. No, because if I get it, then I have to stop taking my medication. Then the back starts to hurt more or the knees I don’t otherwise feel on a daily basis, then they suddenly hurt, and then I can’t actually walk. So it’s not just getting the disease, no, it’s that then I have to stop my medication, maybe 14 days, maybe a month.* (2)

### 3.3. Impact on Everyday Life

The participants who were treated with DMARDs perceived themselves as having increased risk of a severe course of COVID-19 and had added self-imposed precautions to the authorities’ recommendations during the lockdown. Most of them thought they had isolated themselves and their families more than the average population during the pandemic. For some, it led to social deprivation and affected themselves and their families, both emotionally and in practical terms. Daily chores had to be reorganised, e.g., shopping at late hours and tasks “out of the house” were taken over by a family member.

*We are seen—including in the family—as extreme (…) It’s been tough, just having to be inside. That you’ve been locked up (…) And then the thing about your kids, well, can’t we just make a playdate? No, it’s not going to work, you can’t get out there and play with somebody, because we don’t know how careful they are.* (17)

Some participants continued to work from home, even though their colleagues had returned to the office, and this made them feel excluded. Living alone reinforced the experience of isolation, and the lack of social events in particular could lead to frustrations.

One particular consequence of the lock down was the fact that physiotherapists’ services and all training and fitness facilities had been suspended. This had a major impact on the participants’ wellbeing. In particular, the hot-water training was dearly missed. Even if the participants were equipped with home training programmes, they missed the social dimension of group-based training. Some even realised that physical inactivity had led to loss of skills and physical functions.

*I’ve actually felt it on my arms … I’ve never been able to reach into cabinets and stuff like that, but I can tell you, now I can barely even reach the bottom shelves anymore because you didn’t use yourself the same way. And I know it’s my own fault, but how much do you want to sit at home in the living room and start doing exercises. It’s not the same as being around people.* (6)

Another participant said:

*Last fall, I had to buy a walker, and I attribute that to the Corona. Well, I didn’t get enough movement (…) that I wasn’t active enough, you might say, or something like that.* (9)

One participant experienced that not having to spend time commuting to work for an hour every day affected her life in a good way. Another participant lost weight during lockdown.

*For my part, it’s actually been good (…) that’s one of the reasons I’ve lost weight, because I’ve stayed at home and studied online (…) That way I could train and go for long walks, all that kind of stuff.* (19)

Thus, the lockdown turned out to have a positive impact on lifestyle and wellbeing for some.

### 3.4. Position in Society and the Vaccination Program

The vaccination programme gave rise to reflections on whether or not the IA diagnosis was a reason for concern, especially among the participants who still perceived themselves to be at particular risk, even though they were no longer considered to be at high risk by the health authorities, as people with IA were to follow the vaccination schedule for their age group. They felt unsafe and did not understand the changed priorities. They could not let go of the interpretation of being vulnerable.

*(…) now I’m not special. Now I’m just an ordinary citizen (…) It makes you uncomfortable, because you think; someone has assessed that we were in a risk group. And if you just follow the social debate, then you could hear that there was no factual justification for not being in it. It was simply about Denmark not getting enough vaccines, so they had to prioritize them differently.* (2)

The participants who responded the strongest to the changes in the vaccination programme expressed disappointment and a feeling of being worth less than others.

*So you actually react and get upset that you’re not, how am I supposed to express it, not worth more or that… that’s not enough for one to be allowed to get vaccinated. I had a period of that, but luckily, it’s passed.* (7)

The vaccine had also sparked speculation among some participants as to whether it could cause an exacerbation of their rheumatic disease and whether the vaccine would have the same effect if one was treated with immunosuppressive DMARDs. This is illustrated in the following quotation from one of the male participants:

*So what if I get that vaccine? Does it affect my arthritis? Do I get very sick or what does it do?* (3)

### 3.5. Reopening Is Somehow Harder Than Lockdown

In particular, the reopening of society caused reflections among several participants. Some were still nervous and did not feel ready to re-enter society. They were comfortable with the social distance requirements, which justified them maintaining their distance from other, potentially contagious, people. Several participants expressed that they missed the social spirit among Danes during the lockdown and the feeling of being “in this together,” as opposed to the feeling of being left alone with special, and perhaps also self-initiated, limitations as a chronically ill person, while everyone else went back to normal.

*When reopening began. I think that’s been the hardest part. It wasn’t that now we’re not going to do anything. It was starting to be social again.* (4)

*I love the social distance requirement and I’m sorry it’s being removed. But it has also made me isolate myself very, very much.* (18)

In general, regardless of age and social status, the participants were more comfortable with lockdown than the process of reopening, with its much fewer restrictions. This was illustrated by a female participant living alone:

*It is actually the time that’s been hardest for me. Because as long as we were all just shut down; we were all working from home, none of us could participate in leisure activities, none of us could go to the movies, none of us could do anything. There we were in the same boat. But it was hard when the reopening started. Because then, again, you experience that society opens around you. I’m left on the platform, but I can’t just hop on the train, can’t go anywhere, that is … because I’m just not quite ready to jump into the community like everyone else …* (2)

## 4. Discussion

The aim of this study was to explore how people with IA experienced restrictions during the COVID-19 pandemic and their thoughts about the possible impact of the SARS-CoV-2 vaccination on their protection against COVID-19 infection and their everyday lives. The study was performed during a phase where Danish society was characterised by the ongoing transition from a locked-down to a gradually more open society. There was still limited availability of SARS-CoV-2 vaccines, and they were offered to Danish citizens based on the most recent guidance regarding estimates of risk.

We found that, in particular, participants who were treated with DMARDs felt worried and tended to isolate themselves and their families more. Several studies have shown increased levels of worry and lower quality of life among people with chronic conditions [11,13,23,24,25]. In a Danish study, Glintborg et al. documented high levels of anxiety and self-isolation among Danish citizens with IA [15], even though Denmark was found to be among the countries with the lowest level of concern and anxiety among the general population throughout the pandemic [26]. The Danish government and the health authorities aimed for a high level of general information about COVID-19 and restrictions, disseminated through various channels, for example, press conferences [27]. Data from the HOPE-project [28] show that, in general, Danish citizens trusted the government’s and the authorities’ handling of the pandemic. Still, people with IA in our study experienced a lack of targeted and adequate information about their specific vulnerability in relation to risk and consequences of both COVID-19 and the upcoming vaccination. Other European studies have also found that people with chronic conditions experienced both an overload of general information and a lack of targeted and adequate information related to the possible impact of a COVID-19 infection in their specific situation [23,29]. A cross-sectional study across seven countries in central Europe aimed to assess the impact of the pandemic on patients with rheumatic and musculoskeletal diseases [24]. The study concluded that information on COVID-19 had not reached the patients appropriately.

Most participants in our study explained how they felt affected by the inconsistent information throughout the pandemic. They were not able to distance themselves from the interpretation, at the beginning of the pandemic, of being at increased risk due to their chronic illness. Their struggle to gather, understand, and use health information can be associated with the concept of “health literacy.” Health literacy is defined by Liu et al. as “the ability of an individual to obtain and translate knowledge and information in order to maintain and improve health in a way that is appropriate to the individual and system contexts” [30]. Nutbeam proposed that health literacy is needed on both the functional, interactive, and critical levels [31]. It is clear that the complexity of the pandemic and the rapidly emerging evidence required individual health-literacy competence and adaptability among the participants to manage the massive amounts of and continuously changing information, and to select and interpret relevant information and make informed choices considering the risks and consequences that COVID-19 could have for them.

Our study shows that such a lack of knowledge about the possible risks could negatively affect participants’ everyday lives, and thus also their quality of life. Likewise, in a meta-synthesis, Barbara Paterson proposed that the “measure of wellness is determined by comparing the experience to what is known and understood about illness and vice versa” [32]. Paterson’s model, The Shifting Perspectives, describes how living with a chronic disease shifts between the “illness-in-the-foreground” perspective and the “wellness-in-the-foreground” perspective, and that each perspective represents a specific function in a person’s life [32]. Our analysis showed that the insecurity caused by the pandemic and speculations about the impact of the vaccination meant that their chronic illness, which was usually in the background of their everyday lives, suddenly needed more attention. Due to uncertainty about risk, most of our participants reported that IA took on a different and more prominent role in their everyday lives than usual. In light of Paterson’s model, the mere fact of not understanding or knowing the actual risk could be interpreted as a threat to control. Consequently, wellness could deteriorate, and hence negatively affect, quality of life. This may explain why the participants tried to control the situation based on their own interpretation by taking additional self-imposed precautions, especially shielding and self-isolation, to protect themselves and their relatives until they could receive the vaccination. This finding is supported by a Danish national survey that stated that patients with IA isolated themselves more than others in the same age group and that female gender, comorbidities, not working, lower education, biological treatment, and poor quality of life were associated with both anxiety and self-isolation [14]. In accordance with this, we found in our present study that contextual factors, such as the person’s work situation, e.g., whether it was possible to work from home or not, or whether it was possible to continue work as usual; family situation; living alone; having children at home; etc.; influenced the participants’ interpretation of risk.

The Danish sociologist Birthe Bech-Jørgensen argues that all people live different everyday lives, although it always happens in relation to both societal, relational, and individual conditions that provide different opportunities or limitations on how people handle their lives. She emphasizes that we all create our everyday lives through how we handle its conditions [33]. The three predominant societal conditions at stake for the participants in our study were lockdown, reopening, and the vaccination programme. For most of the participants, the reopening of society turned out to be harder to manage than the lockdown, during which they felt on equal footing with other people. The societal conditions also challenged and affected the relational and individual conditions. This is in accordance with what Stine BB Jefsen, a young woman with an IA diagnosis and the winner of the 2021 EULAR Edgar Stene Prize, wrote in her essay “On an equal footing” [34]: “Upon reopening we were no longer in the ‘same boat’ as we were during lockdown.” She described how digital solutions, such as streaming technology, allowed her to attend a concert in her living room and how it made a difference for her during lockdown: “I no longer feel alone. For the first time in a very long time, I feel included and part of a community…” In accordance with some of the findings in our study, she described how the lockdown also had a positive impact on her. She explained how Zoom meetings and online shopping for groceries had made her life with arthritis much easier. Some of the participants in our study encountered a lack of understanding of their self-imposed restrictions in their social network and in society in general. The fact that several participants were very pleased with the social distance requirement and would prefer it to continue after the pandemic also indicates that concerns about infections are not only related to the pandemic but are something these people are concerned about in general, in their everyday lives—with or without the COVID-19 pandemic.

Several participants noted that, even though they had not experienced direct health consequences in terms of infections, severe illness, hospitalization, or delayed follow-up, the pandemic had had indirect consequences for them, both psychologically and physically, mainly because of the lack of physical training and lack of social contact. The study by Mancuso et al. [11] supports the notion that patients with IA reported that coping with challenges due to the pandemic both directly and indirectly worsened their rheumatic symptoms. They also found that, in particular, fatigue and worries about their job and finances were common complaints, although we did not find this in our data. A significant proportion of the participants in our study experienced a decrease in physical activity due to closed training facilities, hot-water pools, and physiotherapist clinics. They found it difficult and demotivating to exercise alone at home. A reduction in physical activity among IA patients may cause physical limitations, increased pain, fatigue, and psychological distress in the short run, and some participants were already experiencing a limitation in motor skills and activity performance due to a lower activity level during lockdown. In a prospective study among 338 British people diagnosed with IA, Sweeney et al. reported that lockdown was associated with worse disease outcomes among the participants. Moreover, they found that physical activity seemed to mitigate both pain, fatigue, and emotional distress [13]. Lévy-Weil et al. found that among 204 patients diagnosed with RA in France, 50% significantly decreased their level of physical activity and 30% reported disease flares during lockdown [25]. Nevertheless, other participants in our study experienced that more time at home and fewer social obligations during the pandemic created the opportunity for more exercise and improved sleep. These findings are supported by Leese et al., who described how some patients with RA were able to improve their self-care while spending more time at home [35].

It is important to note that our findings must be viewed in their temporal context. At the time of the interviews, the participants had just received the first dose of the COVID-19 vaccine, and two of the participants had not accepted the vaccination. Only a few participants questioned the efficacy of the vaccine when taking immune suppressive drugs, and the majority of the participants were simply looking forward to becoming fully vaccinated and recognised the vaccination programme as an important strategy for the whole population. As a consequence of the health authorities’ changed strategies, the participants were offered the vaccines two to three months later than they first expected. The fact that society reopened before they were vaccinated contributed to renewed anxiety among the participants. These concerns may have been influenced by the high levels of uncertainty about the vaccine in society and in the media in general at that time.

### Strengths and Limitations

The participants were all recruited through the DECODIR study, which investigated whether people with IA form antibodies to SARS-CoV-2 at the same level as others. This might have affected their concerns about risk and the effectiveness of the vaccines.

We were not able to conduct face-to-face interviews, and used the phone or an online video platform because of the restrictions in place. Originally, we considered telephone and online interviews to be a limitation because of the inability to read body language and facial expressions. However, despite these limitations, they proved to be a strength, because we were able to secure a safe and flexible setting where the participants did not need to fear infection [36]. Moreover, the distance and anonymity seemed to contribute to a feeling of safety for some respondents, whereas it might have weakened the dialogue for others. Given this uncertainty, we consider that a sample size of 19 was appropriate to reach information power and fulfil the aim of the study [37,38].

It is considered a strength that all interviews were conducted by three of the researchers and that the data were analysed in collaboration and subsequently discussed in the whole research team, which also included the two patient research partners. We consider that this approach strengthened the validity of the findings. The active involvement of all team members represented a variety of perspectives in the phase of data analysis and interpretation and enhanced the validity of the study [39].

## 5. Conclusions

People with IA experienced a lack of consistent information during the pandemic and felt alone in assessing their own risk. For those who perceived themselves to be vulnerable, concerns included whether there was a higher incidence of COVID-19 among people with IA, whether they would be more affected by the virus than others, whether there could be repercussions, and whether the virus could aggravate the rheumatic disease, as well as worries about pausing medication if they became infected.

The COVID-19 pandemic affected the everyday lives of people with IA. Some felt anxiety and imposed more restrictions on themselves than the official recommendations, especially isolation. Some took advantage of the lockdown to exercise more and live more healthily, whereas others experienced loss of skills and physical dysfunctions because of closed training facilities.

People with IA experienced a lack of targeted information about both the vaccine itself and the vaccination programme. The reopening of society gave rise to reflections on societal spirit, the feeling of being in this together, as opposed to the feeling of being left alone with their particular—and perhaps also self-initiated—restrictions as chronically ill.

The findings reveal opportunities for further research on how to support people with IA with relevant and targeted information and support.

## Figures and Tables

**Table 1 vaccines-10-00982-t001:** Interview guide.

What has the COVID-19 pandemic meant to you? What are your thoughts about the risk of you being infected with COVID-19? What medication/treatment do you take for your arthritis? What are your considerations regarding vaccination against COVID-19? What are your thoughts about the present reopening of society?

**Table 2 vaccines-10-00982-t002:** The steps in the inductive qualitative content-analysis process.

Step 1: All audio-recorded interviews, which constituted the unit of analysis, were listened to while reading and adjusting each transcript simultaneously.
Step 2: All transcripts were read again while taking reflective notes and meaning units of importance to the aim of the study were identified.
Step 3: The identified meaning units were condensed and coded according to the manifest content.
Step 4: Codes were grouped into categories based on differences and similarities. This abstraction process was performed in collaboration with a co-researcher.
Step 5: Authors 1 and 2 arranged a workshop for the two patient research partners. They were introduced to a broad range of meaning units within all categories followed by questions for reflection, e.g., “What is at stake for the participants in this situation?” and “Are there other aspects that can give rise to reflection?” Their reflections were audio recorded and notes were taken along the way to support the analysis process.
Step 6: After interpretation of the categories among all authors, they were condensed into five themes. These themes express the underlying meaning of the material on an interpretative level and, thus, the latent content of the material.

**Table 3 vaccines-10-00982-t003:** Characteristics of the participants.

Participant Number	Sex m/f	Age Group	Living Alone/ with Partner	Working Yes/No	Diagnosis	Medical Treatment	Vaccination Type
1	m	50–59	Alone	No	PSA	cDMARD	Pfizer-BioNTech
2	f	50–59	Alone	Yes	SPA	bDMARD	Pfizer-BioNTech
3	m	50–59	Alone	No	MB	None	Pfizer-BioNTech
4	f	50–59	Partner	Yes	MB	None	Pfizer-BioNTech
5	m	50–59	Partner	Yes	RA	cDMARD	Pfizer-BioNTech
6	f	60–69	Partner	No	RA	bDMARD	Pfizer-BioNTech
7	f	50–59	Partner	Yes	RA	cDMARD	Pfizer-BioNTech
8	f	50–59	Partner	Yes	RA	cDMARD	Pfizer-BioNTech
9	m	60–69	Alone	No	PSA	bDMARD	Pfizer-BioNTech
10	m	60–69	Partner	Yes	RA	bDMARD	Pfizer-BioNTech
11	f	60–69	Partner	No	RA	cDMARD	Pfizer-BioNTech
12	f	50–59	Alone	Yes	RA	cDMARD	None
13	f	20–29	Partner	Yes	JIA	bDMARD	None
14	m	40–49	Partner	No	PSA	None	Pfizer-BioNTech
15	f	40–49	Partner	Yes	RA	None	Pfizer-BioNTech
16	f	40–49	Partner	Yes	JIA	None	Pfizer-BioNTech
17	m	30–39	Partner	Yes	SPA	bDMARD	Moderna
18	f	30–39	Alone	No	SPA	bDMARD	Pfizer-BioNTech
19	f	30–39	Partner	Yes	RA	cDMARD	Moderna

m = male, f = female, PSA = psoriatic arthritis, SpA = spondyloarthropathy, RA = rheumatoid arthritis, JIA = juvenile arthritis, MB = Morbus–Bechterew, cDMARD = conventional disease-modifying anti-rheumatic drug, bDMARD = biological disease-modifying anti-rheumatic drug.

**Table 4 vaccines-10-00982-t004:** Emerging themes.

Theme 1	Theme 2	Theme 3	Theme 4	Theme 5
“Changing and divergent information”	“Individual interpretation of own risk”	“Impact on everyday life”	“Position in society and the vaccination programme”	“Reopening is somehow harder than lock-down”

## Data Availability

According to the Danish data protection law, interview data can only be shared after a written data-sharing agreement. In case of interest, please contact the first author.

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
