# Peer review of "Experiences during the COVID-19 Pandemic among People with Inflammatory Arthritis: “Reopening of Society Is Harder than Lock-Down”—A Qualitative Interview Study"

_vaccines, 2022, doi:10.3390/vaccines10070982_

Round 1

Reviewer 1 Report

I enjoyed reading this article. Pointing out how the population reacted to a new virus that soon became a public health problem, becoming a pandemic.

The present article is about arthritis, but we were asked if the patients had other comorbidities besides arthritis? And if they used drugs for this comorbidity?

In the results is started talking about the total number of participants in this study. However, it is interesting that the number of participants is placed in the materials and methods. How many were contacted, how many agreed to participate in this study and how many did not want to participate.

Author Response

Dear reviewer.

We would like to thank you for your highly positive and constructive feedback. In the following, we address each comment point by point with our response and actions in italics.

  1. Reviewers´ comment: “The present article is about arthritis, but we were asked if the patients had other comorbidities besides arthritis? And if they used drugs for this comorbidity?”

Authors’ response: Unfortunately, we have not asked participants if they had comorbidities, and for this study, we did not have access to any of the participants’ medical journals. 

  1. Reviewers´ comment: “In the results is started talking about the total number of participants in this study. However, it is interesting that the number of participants is placed in the materials and methods. How many were contacted, how many agreed to participate in this study and how many did not want to participate”.

Authors’ response and actions: We agree that it is more relevant to mention the number of invited participants in “materials and methods”. We have therefor changed the text in the top of page 3 to: “We invited 19 patients who were already selected to participate in DECODIR to take part in the present study. They all agreed to participate”.

Reviewer 2 Report

The manuscript relates to exploring how people with IA experienced restrictions during the pandemic and the possible impact of vaccination 24 on their protection against COVID-19 and their everyday lives. This is a good study and the manuscript is appropriately written. However, the I suggest following revisions.

1. Introduction part must mention available treatments (remdesivir, molnupiravir, etc.) and the emerging variants of SARS-CoV-2

2. In Table 3 authors must write the name of the vaccine (Pfizer, Moderna, ETC.) and the anti-arthritic medicine used by the participants.

3. A comparative analysis (vaccinate vs unvaccinated participants) about the response given by the 19 participants must be provided results as well as in discussion.

4. Anti-arthritic medicines may be synthetic or monoclonal antibodies. A discussion about the type of anti-arthritic medicine taken by the participant and the vaccine may be a valuable addition to the manuscript.

5. A graphical/pictorial representation of the results will make the manuscript worthy.

6. Does the results have any relations with the emerging variants of SARS-CoV-2? If yes, then it must be mentioned and discussed.

Author Response

Dear reviewer.

We would like to thank you for your positive and constructive feedback. We have reviewed the article again and have made a few text and spelling corrections according to your feedback. The article has also been through professional English editing by Lorna Campbell from the company, Wordwonder. In the following, we address each comment point by point with our response and actions in italics.

  1. Reviewers´ comment: “Introduction part must mention available treatments (remdesivir, molnupiravir, etc.) and the emerging variants of SARS-CoV-2”.

Authors response and actions: We agree that the SARS-CoV-2 variant at the time of the data collection is appropriate to mention in the Introduction section. We have added “At this time the SARS-CoV-2 delta variant was driving the infection in Denmark” and treatments such as Remdesivir and Molnupiravir were not available in Denmark when we started data collection. We have added the following to the Introduction section: “no medical treatment was available nor had been approved by the Danish Medicines Agency”.

  1. Reviewers´ comment: “In Table 3 authors must write the name of the vaccine (Pfizer, Moderna, ETC.) and the anti-arthritic medicine used by the participants”.

Authors’ response and action: We have added information regarding the Disease Modifying treatment and type of SARS-CoV-2 vaccine the participants received to Table 3 as suggested.

  1. Reviewers´ comment: “A comparative analysis (vaccinate vs unvaccinated participants) about the response given by the 19 participants must be provided results as well as in discussion”.

Authors’ response: We agree that if we had a higher number of participants, a comparative analysis would have added valuable knowledge to the different perspectives regarding experiences during a pandemic. As we only had two participants who had declined the vaccine, we did not have statistical power to include a comparative analysis in the present study.

  1. Reviewers´ comment: Anti-arthritic medicines may be synthetic or monoclonal antibodies. A discussion about the type of anti-arthritic medicine taken by the participant and the vaccine may be a valuable addition to the manuscript.

Authors’ response: We agree that understanding and discussion of the potential impact of type of anti-rheumatic treatment on vaccination could have been of interest. However, it was beyond the objectives of this study to explore this in detail. This was due to the explorative and qualitative nature of the study. Furthermore, detailed information on type of treatment in this small patient cohort would potentially compromise patient anonymity.

  1. Reviewers´ comment: A graphical/pictorial representation of the results will make the manuscript worthy.

Authors response: We have added a table (Table 4) to illustrate the findings:

Table 4: Emerging themes

Theme 1

Theme 2

Theme 3

Theme 4

Theme 5

“Changing and divergent information”

“Individual interpretation of own risk”

“Impact on everyday life”

“Position in society and the vaccination programme”

“Reopening is somehow harder than lock-down”

  1. Reviewers´ comment: Does the results have any relations with the emerging variants of SARS-CoV-2? If yes, then it must be mentioned and discussed.

Authors’ response: It is an interesting perspective. In retrospect, the Delta variant caused more serious disease including more hospitalizations than the Omicron variant. However, it is our belief that it did not have any impact on the results, as this was not known by us or the participants at the time of data collection.